# Hypoxia-induced miR-92a regulates p53 signaling pathway and apoptosis by targeting calcium-sensing receptor in genetically improved farmed tilapia *(Oreochromis niloticus)*

**Jun Qiang●\*●, Jie He●, Yi-Fan Tao, Jin-Wen Bao, Jun-Hao Zhu, Pao Xu\***

Key Laboratory of Freshwater Fisheries and Germplasm Resources Utilization, Ministry of Agriculture, Freshwater Fisheries Research Center, Chinese Academy of Fishery Sciences, Wuxi, Jiangsu, China

● These authors contributed equally to this work.
\* qiangjunn@163.com (JQ); Xup@ffrc.cn (PX)

**Data Availability Statement:** All relevant data are within the manuscript and its Supporting Information files.

## Abstract

miR-92a miRNAs are immune molecules that regulate apoptosis (programmed cell death) during the immune response. Apoptosis helps to maintain the dynamic balance in tissues of fish under hypoxia stress. The aim of this study was to explore the role and potential mechanisms of miR-92a in the liver of tilapia under hypoxia stress. We first confirmed that *CaSR* (encoding a calcium-sensing receptor) is a target gene of miR-92a in genetically improved farmed tilapia (GIFT) using luciferase reporter gene assays. In GIFT under hypoxia stress, miR-92a was up-regulated and *CaSR* was down-regulated in a time-dependent manner. Knocked-down *CaSR* expression led to inhibited expression of *p53*, *TP53INP1*, and *caspase-3/8*, reduced the proportion of apoptotic hepatocytes, and decreased the activity of calcium ions induced by hypoxia in hepatocytes. GIFT injected in the tail vein with an miR-92a agomir showed up-regulation of miR-92a and down-regulation of *CaSR*, *p53*, *TP53INP1*, and *caspase-3/8* genes in the liver, resulting in lower serum aspartate aminotransferase and alanine aminotransferase activities under hypoxia stress. These findings suggest that stimulation of miR-92a interferes with hypoxia-induced apoptosis in hepatocytes of GIFT by targeting *CaSR*, thereby alleviating liver damage. These results provide new insights into the adaptation mechanisms of GIFT to hypoxia stress.

## Introduction

miRNAs are a small class of non-coding RNAs that regulate the expression of one or more target genes by binding to their 3′-untranslated region (UTR). In recent years, the potential application of members of the miR-92 family as immune molecules in aquaculture has received extensive attention [1–4]. Previous studies have shown that the miR-92 family is closely related to the immune response of sea cucumber (*Apostichopus japonicus*) to vibriosis caused by *Vibrio splendidus* [1]. That study showed that miR-92 can regulate the host–pathogen interaction

**Funding:** This work was supported by the Central Public-interest Scientific Institution Basal Research Fund, Chinese Academy of Fishery Sciences (NO. 2018HY-XKQ02-01; 2019ZY19; 2019JBFC01). As the funder of the project, Jun Qiang plays an important role in the study design and preparation of the manuscript.

**Competing interests:** The authors have declared that no competing interests exist

**Abbreviations:** UTR, untranslated region; GIFT, genetically improved farmed tilapia; TP53INP2, p53-inducible nuclear protein 2; CaSR, calcium-sensing receptor; NC, negative control; DO, dissolved oxygen; SDS, sodium dodecyl sulphate; TBST, Tris-buffered saline with Tween; ALT, alanine aminotransferase; AST, aspartate aminotransferase; GAPDH, glyceraldehyde-3-phosphate dehydrogenase.

in sea cucumber by binding to two candidate genes, one encoding polyepidermal growth factor-like domain 6 and the other encoding SMAD-specific E3 ubiquitin protein ligase. In oyster (*Crassostrea gigas*), miR-92d regulates the expression of tumor necrosis factor (TNF) by targeting the coding region of *CgLITAF3*, which encodes lipopolysaccharide-induced TNF-a factor 3, thereby triggering an inflammatory response to invading bacteria [2]. In amphioxus (*Branchiostoma belcheri*), miR-92d regulates the immune response to bacterial infection by binding to the gene encoding complement C3 [3]. In genetically improved farmed tilapia (GIFT), (*Oreochromis niloticus*), miR-92d-3p is involved in mediating the expression of complement C3, and inhibition of miR-92d-3p promotes complement C3 expression and enhances the inflammatory response to *Streptococcus iniae* infection [4].

In a previous study, we successfully constructed miRNA expression libraries from uninfected GIFT and those infected with *S. iniae*. Gene enrichment pathway analyses indicated that one of the differentially expressed miRNAs, miR-92a, may be involved in cell signal transduction or immune regulation processes in GIFT [5]. *CaSR*, which encodes a calcium-sensing receptor, may be a potential target gene of miR-92a (free energy, −26.5 Kcal/mol) in GIFT. Recent studies have found that activated CaSR is involved in the regulation of cellular inflammatory responses and apoptosis signaling [6–8]. In mammals, CaSR is a sensitive receptor for $Ca^{2+}$, and its expression is regulated by the concentration of extracellular $Ca^{2+}$. As a multifunctional regulator, CaSR participates in G protein signal transduction and GPCRs kinase (GPK)-induced desensitization by altering the intracellular $Ca^{2+}$ concentration, thereby mediating processes such as cell growth, differentiation, and ion channel opening [9]. To expand on our previous studies, we aimed to confirm the relationship between miR-92a and its target gene *CaSR*, and determine how this relationship regulates cell stress responses in GIFT. We also aimed to clarify its regulatory pathway and activation mechanism. Addressing these questions will shed light on the molecular mechanisms of stress regulation in fish.

Changes in dissolved oxygen (DO) can affect fish survival. Generally, fish grow and develop normally when the DO level is higher than 4.0 mg/L. Low DO levels (<2.0 mg/L) cause a floating phenomenon in farmed fish. If the DO level drops below 1.0 mg/L (hypoxic conditions), most fish will severely float their heads and eventually suffocate to death [10]. There are many causes of fish death under hypoxia stress, including an imbalance of apoptosis [11]. Apoptosis is the orderly death of cells controlled by genes to maintain the stability of the internal environment. Apoptosis and the immune response involve proteins including p53, p53-inducible nuclear protein 2 (TP53INP2), caspase-3, and caspase-8. Among them, the p53 signaling pathway is considered to be a key factor in the regulation of apoptosis, because it directly or indirectly induces multiple regulatory genes to interfere with apoptosis in fish [12]. As a downstream gene of p53, *TP53INP* encodes a protein that coactivates transcription factors such as p53 in the nucleus to regulate apoptosis and the expression of cell cycle-related genes [13]. Apoptosis-associated caspases are widely expressed in various fish exposed to hypoxia, and their activity leads to impairment of liver metabolism and calcium homeostasis [14, 15]. The expression levels of apoptosis-related genes (*caspase-3/8/9*) were found to be low in the gill and liver of healthy largemouth bass (*Micropterus salmoides*) but increased after exposure to hypoxia [16, 17]. Similarly, Atlantic croaker (*Micropogonias undulatus*) under hypoxia stress showed increased caspase-3/7 activity and elevated expression of the proapoptotic genes *Bax* and *p53* in the ovaries, and the apoptotic pathway was activated [18].

Tilapia is one of the most important freshwater farmed fish in China. High temperatures or deterioration of water quality can cause a sharp drop in DO levels in aquaculture ponds, leading to fish death [19]. Therefore, the main objectives of this study were as follows: 1) to verify that *CaSR* is a potential target gene of miR-92a; 2) to determine the effect of *CaSR* knockdown on the $Ca^{2+}$ concentration and apoptosis pathway in GIFT hepatocytes; and 3) to explore how

miR-92a expression is activated in GIFT under hypoxia stress. The results of this study provide new information about stress regulation and adaptation mechanisms in fish.

## Materials and methods

### Ethics statement

The study protocols and design were approved by the Ethics Committee at the Freshwater Fisheries Research Center of the Chinese Academy of Fishery Sciences (Wuxi, China). The Ethics approval number for this experiment is 2019–028. The GIFT were maintained in well-aerated water and treated with 200 mg/L tricaine methane sulfonate (Sigma, St Louis, MO, USA) for rapid deep anesthesia. The samples were extracted based on the Guide for the Care and Use of Laboratory Animals in China.

### Analysis of regulatory relationship between miRNA and its target genes

We randomly assigned 180 healthy GIFT (average fish weight: 18.4 g ± 0.8 g) to nine experimental groups (20 fish/group). There were three treatment groups: an miR-92a agomir group; an miR-92a negative agomir group; and a control group treated with phosphate-buffered saline (PBS). Each group had three replicates of 20 fish. The chemically synthesized miRNA agomir (5′-UUGCACUUGUCCCGGCCUGU-3′) and miRNA negative agomir (5′-UUUUUUAAGUC CCGGCCUGU-3′: with a six mismatch mutation in its complementary bases) were designed and synthesized by the Ruibo Biological Technology Co., Ltd. (Guangzhou, China). The miRNA negative agomir was administered by tail vein injection at a dose of 25 mg/kg body weight. The same dose of negative agomir or PBS was administered to fish in the negative agomir or PBS groups, respectively, in the same way. At 0 h, 12 h, 24 h, and 48 h post-injection, three fish were randomly selected from each tank and deeply anesthetized before dissecting liver tissues. The liver samples were snap frozen in liquid nitrogen, and then stored at −80°C until analyses of gene expression and protein levels.

### Identification of binding sites of miR-92a-CaSR 3′ UTR

The full-length *CaSR* 3′ UTR sequence (XM_025910710.1) of GIFT was synthesized and then inserted into the pGL3-control vector, yielding 3′ UTR wild-type. In the seed sequence region where miRNA binds to its target gene, the six complementary bases (AAUUGU) of CaSR 3′-UTR were replaced with GGCAAG to construct the 3′UTR mutant. The procedures for HEK293T cell culture, vector transfection, and luciferase activity determination were as described previously [18]. We used renilla luciferase activity to standardize the luciferase activity of each transfected cell. As the miRNA negative control (NC), we selected the miRNA of *Caenorhabditis elegans* (5′-UUUGUACUACACAAAAGUACUG-3′), which has minimal homology with all miRNAs in the miRBase database (http://www.mirbase.org/). There were four treatment groups in this experiment: CaSR 3′ UTR-wild type + miRNA mimic; CaSR 3′ UTR-Mutant + miRNA mimic; miRNA NC (Ctrl mimic) + CaSR 3′ UTR-wild type; Ctrl mimic + CaSR 3′ UTR-Mutant. Each treatment had eight replicates. Luciferase activity was detected to determine the binding efficiency of miRNA to its target gene.

### Regulatory response of GIFT hepatocytes under hypoxia stress after knocking down expression of *CaSR*

We constructed an RNA interference (RNAi) expression vector to knock down the expression of *CaSR* in GIFT. First, according to the GIFT *CaSR* gene sequence, we designed an RNAi sequence (5′-GAGAGCACAGATGACTGGTGATATT-3′) with no homology to other coding

sequences in tilapia. The construction of the positive interference plasmid (CaSR knockdown), the negative control interference plasmid (NC), and the control plasmid (Con) were as described in Qiang et al. (2020). Hepatocytes were cultured, isolated, and purified as described by Chen et al. (2011). This experiment had four experimental groups of cells: three groups were subjected to a hypoxia treatment (CaSR knockdown group, NC group, and Con group), and one group, the normal group (NG), was not subjected to a hypoxia treatment. Each treatment had eight replicates. The primary cultured hepatocytes were grown to about 80%–90%, and then transfected using Lipofectamine 2000. The three hypoxia-treated cell groups were placed in an YQX-II anoxic incubator (JTONE, Hangzhou, China) (27°C, $N_2$ 93%, $O_2$ 2%, $CO_2$ 5%) for 24 h to impose hypoxia stress. The cell lysate was treated for 30 min, and then centrifuged (4°C, 12000 $g$, 20 min). The supernatant was collected for further analyses. The hepatocytes in the NG were placed in an Herocell 180 incubator (RADOBIO, Shanghai, China) and cultured under normoxic conditions (27°C, $O_2$ 20%, $CO_2$ 5%) without transfection and hypoxia stress.

## miR-92a-mediated stress response in GIFT liver after hypoxia exposure

The experimental design was based on that of Qiang et al. [20]. In total, 360 GIFT juveniles (average size, 26.8 g ± 0.9 g) were randomly assigned to nine 400-L tanks (40 fish/tank). The synthesis, dissolution, and injection of the miR-92a agomir (5′–UUGCACUUGUCCCGGCCU GU–3′) and miRNA NC (5′–UUUGUACUACACAAAAGUACUG–3′) were as described in the section "Analysis of the regulatory relationship between miRNA and its target genes". The control group was similarly injected with PBS. The experimental GIFT at 12 h post-injection were subjected to low-DO (1.0 mg/L) stress for 96 h. The DO level in water was determined using an LDO101 probe (Hach, Loveland, CO, USA). Nitrogen and air charges were used to regulate and maintain the DO level in water. Three fish were randomly selected from each group at each sampling time point, and blood was drawn and liver tissues were dissected quickly. The liver samples were snap-frozen in liquid nitrogen, and then stored at −80°C until analyses of mRNA levels. Blood samples were left at 4°C for 2 h and then centrifuged (4°C, 3500 $g$, 10 min) to collect the serum. The serum was kept at −80°C until analysis.

## Measurement indices

**mRNA expression.** Total RNA extraction, reverse transcription (RT), and quantitative real-time PCR (qRT-PCR) were conducted as described elsewhere [21]. The qRT-PCR analyses were conducted using the ABI QuantStudio 5 Real-Time PCR System (Foster City, CA, USA) with gene-specific primers (see Table 1). The endogenous control gene was *18S rRNA*.

**miRNA expression.** The RT reaction and qRT-PCR of the miRNA were as described elsewhere [21]. The primers for miR-92a are shown in Table 1. The reference gene was *U6*.

**Western blot analysis.** Each liver tissue sample (0.1 g) was ground in liquid nitrogen with a mortar and pestle, then 1 mL ristocetin-induced platelet agglutination buffer (containing 1% v/v 10 mg/ml phenylmethanesulfonyl fluoride) was added and the mixture was homogenized (4°C, 15,000 g, 1 min) using a Polytron (PT2500E, Kinematica, Lucerne, Switzerland). The procedures for SDS-PAGE preparation, protein sample electrophoresis, membrane transfer, blocking, and antibody incubation were as described by Qiang et al. [20]. Color was developed using Immobilon Western HRP substrate (Millipore, Billerica, MA, USA). Glyceraldehyde-3-phosphate dehydrogenase (GAPDH) was the reference protein.

**Identification of apoptotic cells and determination of $Ca^{2+}$ concentration.** The culture, collection, and apoptotic cell detection of hepatocytes were conducted as described by Qiang et al. [21]. Hepatocytes were collected and resuspended in pre-cooled PBS, and then the $Ca^{2+}$-fluorescent

**Table 1. Sequences of primers used for qRT-PCR.**

| Name | Primer sequence (5'-3') |
|---|---|
| CaSR | F: 5'-AAAATCTATGATGCTTGTGGCTCC-3'<br>R: 3'-ATTGCCATGCAAGGGGAG-5' |
| TP53INP2 | F: 5'- TACGCGCAGCAATGTTTCAG-3'<br>R: 5'- GGCTCCCTCAGGCAGATTGA-3' |
| p53 | F: 5'-TTTTCTCCTCCCTGTTCGTGG-3'<br>R: 5'- CGGGAACCTCATGCTTCACT-3' |
| Caspase-3 | F: 5'- GAAACGAACAGCAGCAGACC-3'<br>R: 5'- CGAGTGCTCATCCCTGTTGT-3' |
| Caspase-8 | F: 5'- ACAGACAAAGGGCTCGTCAG-3'<br>R: 5'- GAAGACCACGCACAAACCAC-3' |
| 18S rRNA | F: 5'-GGCCGTTCTTAGTTGGTGGA-3'<br>R: 5'-TTGCTCAATCTCGTGTGGCT-3' |
| miR-92a | 5'- AAGCGACCTATTGCACTTGTCC-3' |

probe Fluo-3AM was added to a final concentration of 5 μg/mL. The mixture was incubated in the dark for 40 min at room temperature and then passed through a cell sieve (40 μm mesh size). The $Ca^{2+}$ concentration was detected using the Incyte module of a FACSCalibur flow cytometer with excitation at 506 nm and emission at 526 nm.

**Measurement of activities of aspartate aminotransferase and alanine aminotransferase in serum.** The activities of aspartate aminotransferase (AST) and alanine aminotransferase (ALT) in serum were determined using kits obtained from the Jiancheng Bio-Engineering Institute (Nanjing, China), according to the manufacturer's instructions. Absorbance at the wavelengths specified in the kit manuals was determined using a BioTek Eon Microplate Spectrophotometer (BioTek, Winooski, VT, USA).

## Data processing and statistical analysis

The relative expression levels of mRNA and miRNA were calculated using the $2^{-\Delta\Delta CT}$ method. Data from all experiments were statistically analyzed using SPSS 21.0 software. Shapiro–Wilk's test was used to test the normal distribution of data ($\alpha = 0.1$). Levene's test was used to determine the homogeneity of variance ($\alpha = 0.1$). The experimental data shown in figures and tables are mean ± standard deviation. When the experimental data met the tests for normality and homogeneity of variance, we performed analysis of variance and selected appropriate statistical methods for comparisons among treatment groups. Independent samples t test was used to compare different treatment times within the same experimental group; Duncan's multiple comparison was used to compare different treatment groups at the same time. Differences were considered significant at $P<0.05$. Data that were not normally distributed were log-transformed before statistical analysis.

## Results

### miR-92a binds to *CaSR* 3'-UTR to regulate mRNA expression

To analyze the regulatory relationship between miR-92a and its target gene, we injected the miR-92a agomir (negative agomir or PBS (control) into the tail vein of GIFT). The levels of hepatic *CaSR* mRNA were significantly lower in GIFT treated with the miRNA agomir than in the control group at 12, 24, and 48 h after injection (Fig 1A, $P < 0.05$). However, at 12, 24, and 48 h after injection, the levels of miR-92a were significantly higher in the miRNA agomir group than in the control group and the negative agomir group (Fig 1B, $P < 0.05$). The CaSR

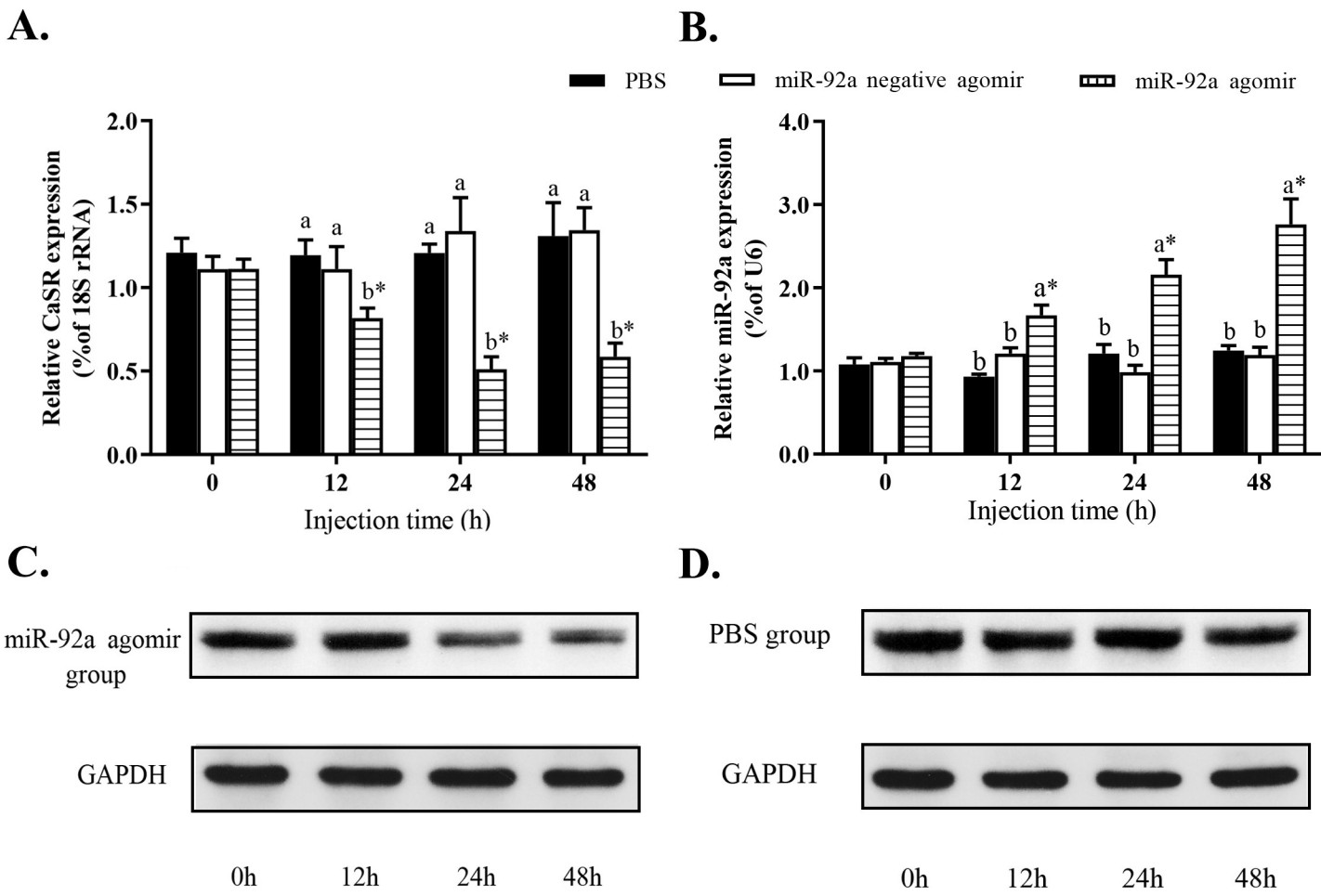

**Fig 1.** Analysis of regulatory relationship between *CaSR* (A) and miR-92a (B) *in vivo*. Juvenile GIFT weighing 18.4 g ± 0.8 g were injected in tail vein with miR-92a agomir (dose, 25 mg/kg body weight), miR-92a negative agomir, or PBS (control). (A, C, and D) qRT-PCR (upper panel) and western blot (lower panel) analyses of CaSR expression in GIFT after injection with miR-92a agomir. Bands of interest were cropped from the same membrane. (B) qRT-PCR analysis of miR-92a expression in GIFT after injection with miR-92a agomir. Based on relative levels in control group at 0 h, relative expression levels of mRNA and miRNA in each experimental group were determined by $2^{-\Delta\Delta CT}$ method. * Indicates significant difference between pre- and post- injection in the same experimental group (Independent samples t test; $P < 0.05$). Different lowercase letters indicate significant differences among different treatments at each sampling point (Duncan's multiple range test; $P < 0.05$).

protein level was significantly lower in the miRNA agomir group (Fig 1C and S1–S3 Figs, $P < 0.05$) than in the control group at 24 h and 48 h after injection (Fig 1D). There was no significant difference in the levels of *CaSR* mRNA and miR-92a between the control group and the negative agomir group at each time point (Fig 1A and 1B, $P > 0.05$). These results showed that up-regulation of miR-92a expression led to down-regulation of *CaSR*.

We generated an miR-92a mimic to target the 3′ end of *CaSR*, and then tested its efficacy by conducting dual luciferase reporter assays using HEK293T cells. Luciferase activity in the *CaSR* 3′ UTR wild type + control mimic group (Ctrl group, no mimic) was 1.12 ± 0.03, while that in the UTR wild type + miR-92a mimic group was decreased by 48.47% (Fig 2A) (statistical significance; t = 2.357, $P = 0.023$). To confirm that binding between the *CaSR* 3′ UTR and the miR-92a sequence was essential for this activity, we constructed a mutated *CaSR* 3′ UTR sequence (3′ UTR Mut). The luciferase activity levels in the 3′ UTR Mut + Ctrl mimic group or the miR-92a mimic group were 0.96-times and 0.87-times that in the control group, respectively; these differences were not significant ($P > 0.05$). These results confirmed that GIFT miR-92a binds to the 3′ UTR of *CaSR* mRNA. Although the 5′ end of the miR-92a sequence

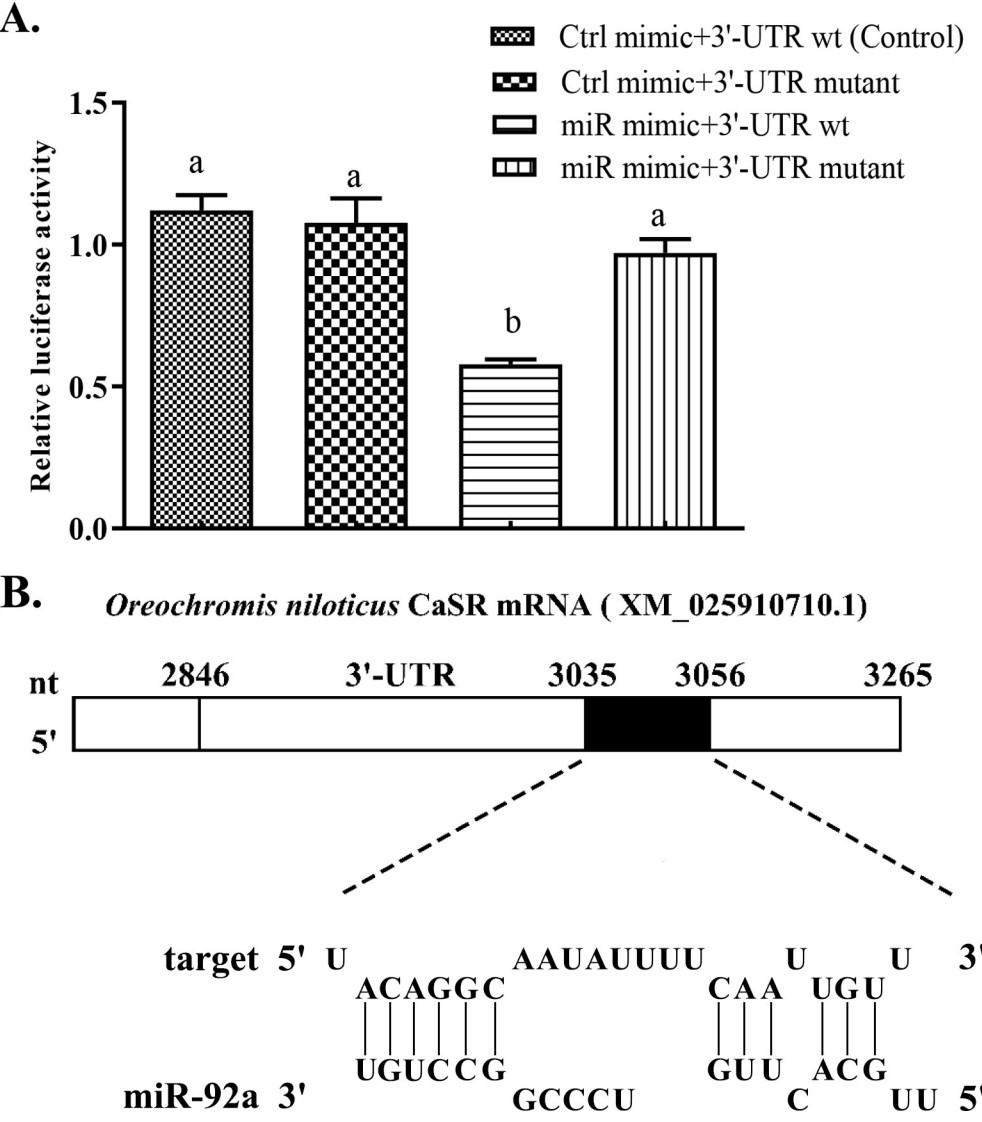

**Fig 2. Verification of binding between miRNA and its potential target gene.** (A) Validation of miRNA binding to 3′ UTR of potential target gene using dual luciferase reporter system. HEK-293T cells in 12-well plates were co-transformed with pGL-CaSR 3′ UTR (3′ UTR-wild type) or pGL- CaSR (3′ UTR-Mutant) and miRNA mimic (miR mimic) or miRNA negative control (Ctrl mimic) using Lipofectamine 2000 transfection reagent. Luciferase activity was determined based on fluorescence. (B) Analysis of binding site between miR-92a and *Oreochromis niloticus CaSR* (XM_025910710.1): 5′-end of miR-92a can pair with 3048 bp–3056 bp position at the *CaSR* 3'-UTR. Compensation binding site at 3′-end of miR-92a enhances its binding stability with 3′-UTR of *CaSR*. Different lowercase letters indicate significant differences among experimental groups (Duncan's multiple comparison; $P < 0.05$).

showed only partial complementarity in a 2–8 bp region, a compensation site at the 3′ end enhanced its binding stability to the 3′-UTR of the target *CaSR* mRNA (Fig 2B).

## CaSR knockdown regulates Ca$^{2+}$ concentration and expression of genes related to cell signaling pathway in GIFT hepatocytes

Next, we determined the effects of CaSR knockdown on the responses of GIFT hepatocytes under hypoxia stress. This experiment consisted of four groups of GIFT hepatocytes: a normal group (NG, no transfection, no hypoxia); a control + hypoxia stress group (Con + hypoxia); a

CaSR-negative control transfection + hypoxia stress group (NC + hypoxia); and a CaSR knock-down + hypoxia stress group (Knockdown + hypoxia).

As shown in Fig 3A, compared with the *CaSR* transcript levels in the Con + hypoxia group and the NC + hypoxia group, that in the Knockdown + hypoxia group was decreased by 60% ($P < 0.05$). The transcript levels of *TP53INP2* (Fig 3B), *p53* (Fig 3C), *caspase-3* (Fig 3D), and *caspase-8* (Fig 3E) were significantly lower in the Knockdown+ hypoxia group than in the Con + hypoxia and NC + hypoxia groups. The percentage of hepatocytes in the early and late stage of apoptosis was 14.62% in the Con + hypoxia group and 14.28% in the NC + hypoxia group (Q4 +Q2 in Fig 4A), significantly higher than in the NG and Knockdown + hypoxia group (Fig 4A). The Ca$^{2+}$ concentration was significantly lower in the Knockdown + hypoxia group than in the Con + hypoxia and NC + hypoxia groups (Fig 4B), but not significantly different between the NG and the knockdown + hypoxia group ($P>0.05$).

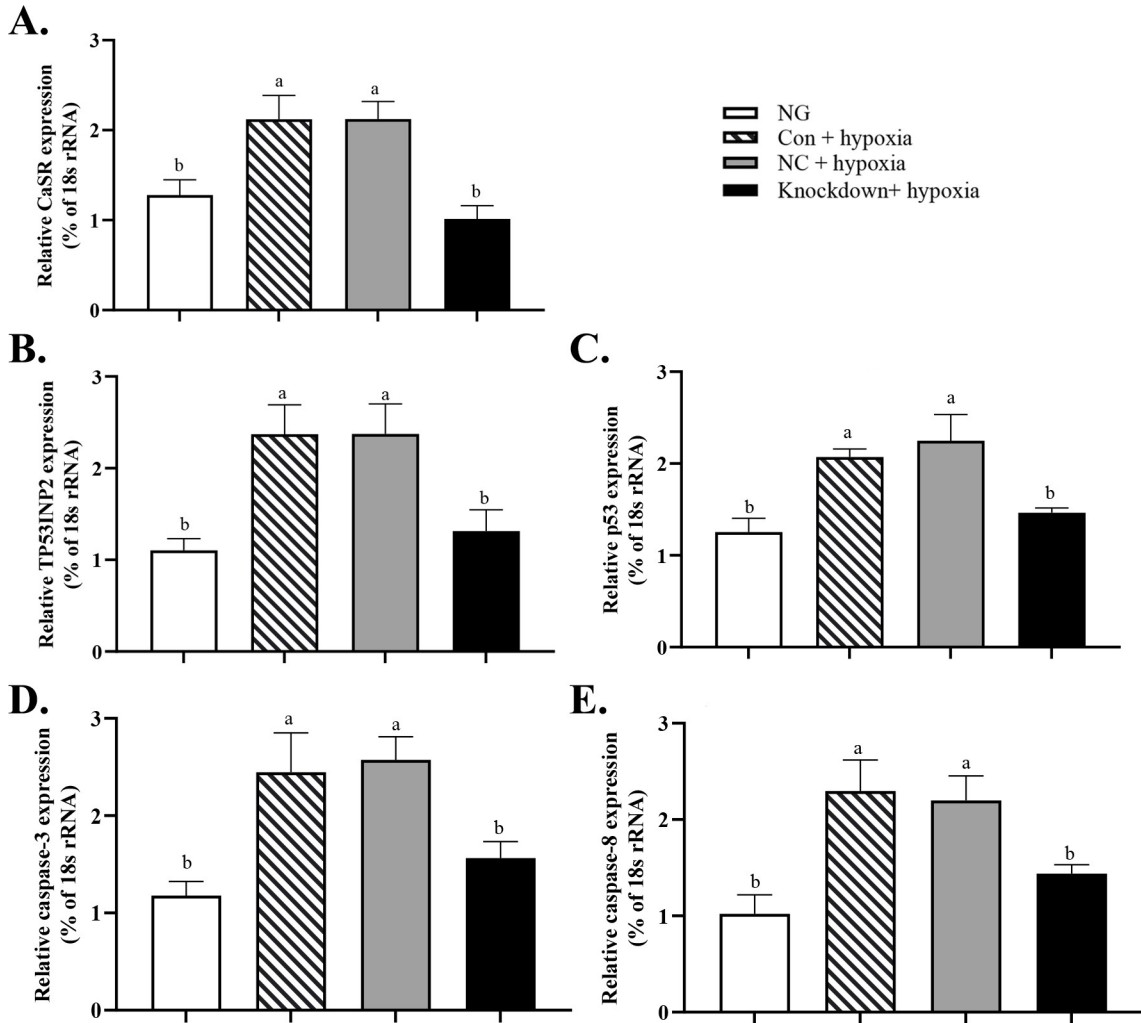

**Fig 3. Effect of *CaSR* knockdown on transcript levels of *CaSR*, *p53*, *p53-inducible nuclear protein 2*, *caspase-3*, and *caspase-8* in GIFT hepatocytes under hypoxia stress.** (A) Effect of *CaSR* knockdown on transcript levels of *CaSR* in GIFT hepatocytes under hypoxia stress. Hepatocytes were transfected with *CaSR* knockdown, NC, or Con vector using Lipofectamine 2000 and subjected to 24 h of hypoxia stress in an incubator (knockdown+ hypoxia; NC+ hypoxia, and Con+ hypoxia, respectively). Normal group (NG): hepatocytes under normoxic conditions (no transfection, no hypoxia). (B, C, D, and E) Effect of *CaSR* knockdown on transcript levels of *p53*, *p53-inducible nuclear protein 2*, *caspase-3* and *caspase-8* in GIFT hepatocytes under hypoxia stress. Based on relative levels of respective mRNAs in NG, relative mRNA levels in each group were determined by 2$^{-\Delta\Delta CT}$ method. Different lowercase letters indicate significant differences among experimental groups (Duncan's multiple range test, $P < 0.05$).

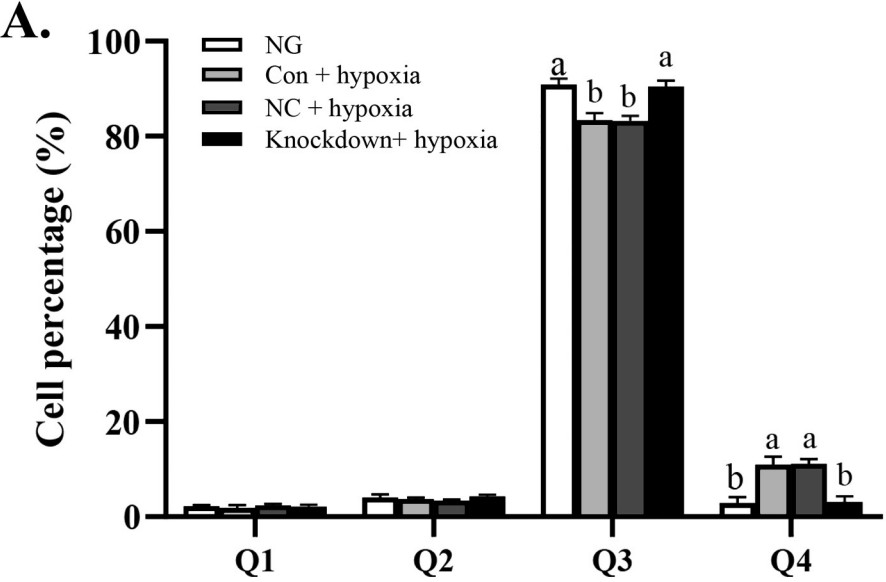

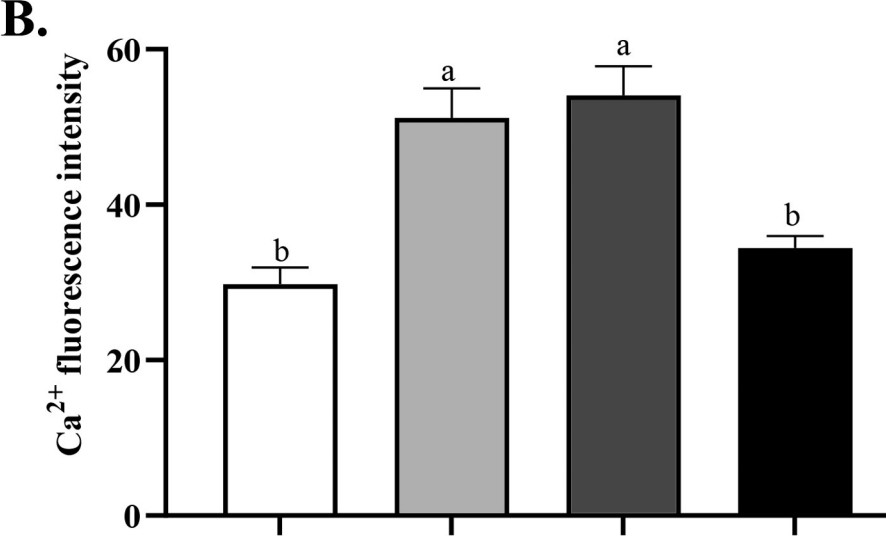

**Fig 4. Effect of *CaSR* knockdown on apoptosis and Ca²⁺ concentration in hepatocytes of GIFT under hypoxia stress.** (A) Proportion of apoptotic cells in NG, knockdown+ hypoxia, NC+ hypoxia and Con+ hypoxia groups. Figure shows percentages of dead cells (Q1), late apoptotic cells (Q2), viable cells (Q3), and early apoptotic cells (Q4). (B) Ca²⁺ fluorescence intensity in NG, knockdown+ hypoxia, NC + hypoxia, and Con + hypoxia groups. Different lowercase letters indicate significant differences among experimental groups (Duncan's multiple range test, $P < 0.05$).

## miR-92a regulates *p53*, *TP53INP2*, *caspase-3/8* and serum AST and ALT activities in GIFT under hypoxia stress

To further analyze the role of miRNA-mediated *CaSR* expression in the GIFT liver in response to hypoxia stress, juvenile GIFT were injected with miR-92a agomir, miRNA NC, or PBS (control) via the tail vein. At 12 h after miRNA agomir injection, the GIFT were subjected to

hypoxia stress. At each time point, the level of miR-92a was significantly higher in the agomir group than in the control and the NC groups (Fig 5A, $P<0.05$). The expression levels of miR-92a in the control and NC groups were significantly lower at 48 h and 72 h of hypoxia stress than before the stress treatment ($P<0.05$). There was no significant difference in the expression level of miR-92a between the control group and the NC group at each sampling time (Fig 5A, $P>0.05$). The transcript level of *CaSR* was significantly lower in the agomir group than in the control group at 12, 24, 48, 72, and 108 h of hypoxia stress (Fig 5B).

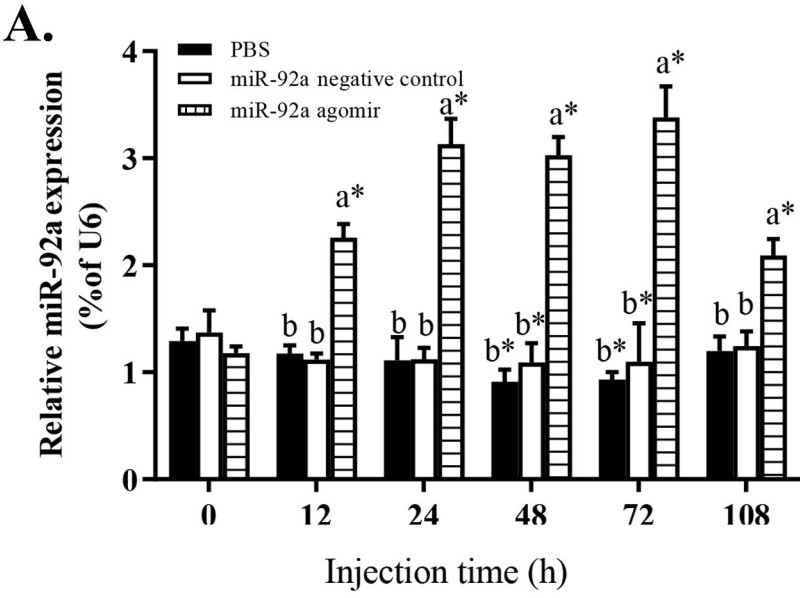

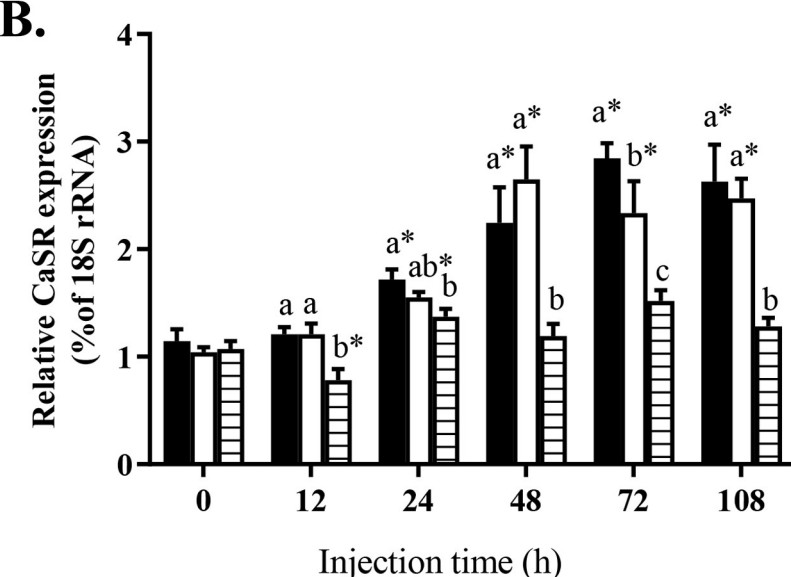

**Fig 5.** Effect of promoting miR-92a expression on transcript levels of miR-92a (A) and *CaSR* (B) in GIFT liver. Juveniles weighing 26.8 g ± 0.9 g were injected in tail vein (dose, 25 mg/kg body weight) with miRNA negative control, or miR-92a agomir, or PBS (control) and response was monitored for 108 h. Based on relative expression levels in control group at 0 h, relative expression levels of mRNA and miRNA in each experimental group were determined by $2^{-\Delta\Delta CT}$ method. * Indicates significant differences between pre- and post- injection in the same experimental group (Independent samples t test; $P < 0.05$). Different lowercase letters indicate significant differences among different treatments at each sampling point (Duncan's multiple comparison; $P < 0.05$).

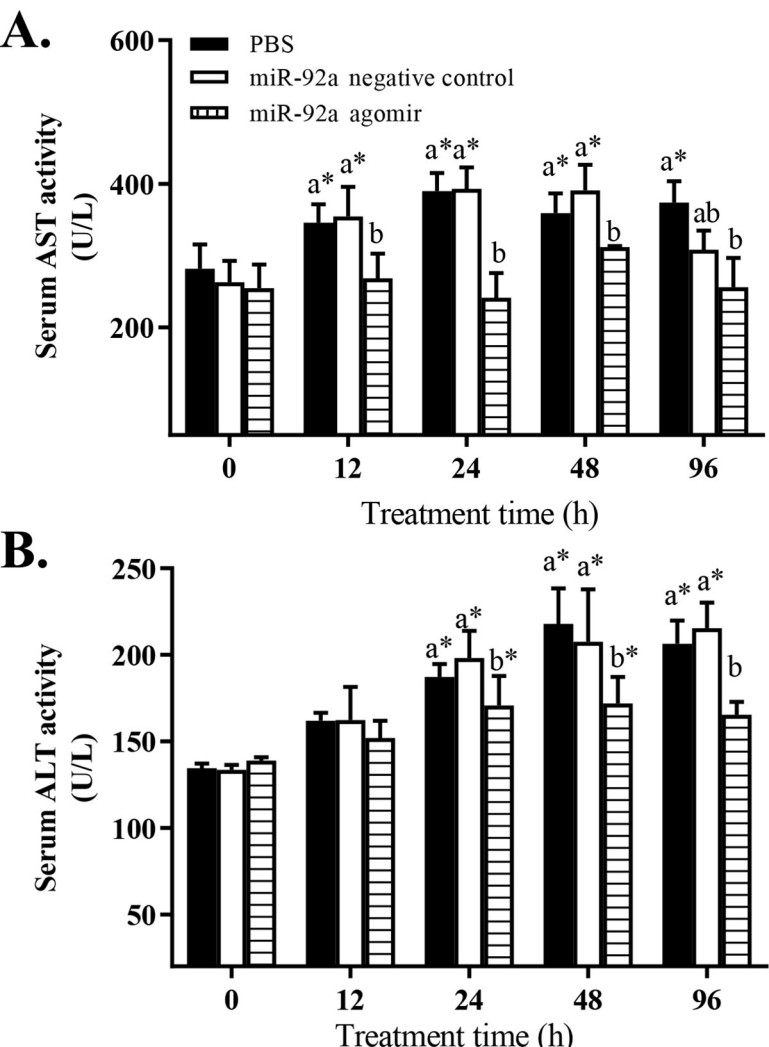

**Fig 6.** Effect of promoting miR-92a expression on serum aspartate aminotransferase (A) and alanine aminotransferase (B) activities in GIFT under hypoxia stress. At 12 h after miR-92a agomir injection or PBS (control), juveniles were subjected to hypoxia stress for 96 h. * Indicates significant differences between of pre- and post- injection in the same experimental group (independent samples t test; $P < 0.05$). Different lowercase letters indicate significant differences among different treatments at each sampling point (Duncan's multiple comparison; $P < 0.05$).

After 24 h of hypoxia stress, the serum AST (Fig 6A) and ALT (Fig 6B) activities were significantly lower in the agomir group than in the control and the NC groups. In all treatment groups, the serum AST and ALT activities tended to increase under hypoxia stress. Serum ALT activity in the agomir group was significantly higher at 24 and 48 h than at 0 h (pre-stress). The transcript levels of *p53*, *TP53INP2*, *caspase-3*, and *caspase-8* in the liver tended to increase under hypoxia stress in the control group and the NC group (Fig 7). The transcript level of *TP53INP2* was significantly lower in the agomir group than in the control group at 48 and 96 h of hypoxia stress (Fig 7A). The transcript levels of *p53* (Fig 7B) and *caspase-8* (Fig 7D) were lower in the agomir group than in the control group at 96 h of hypoxia stress. The transcript levels of *caspase-3* were significantly lower in the agomir group than in the control group and the NC group at 24, 48, and 96 h of hypoxia stress (Fig 7C).

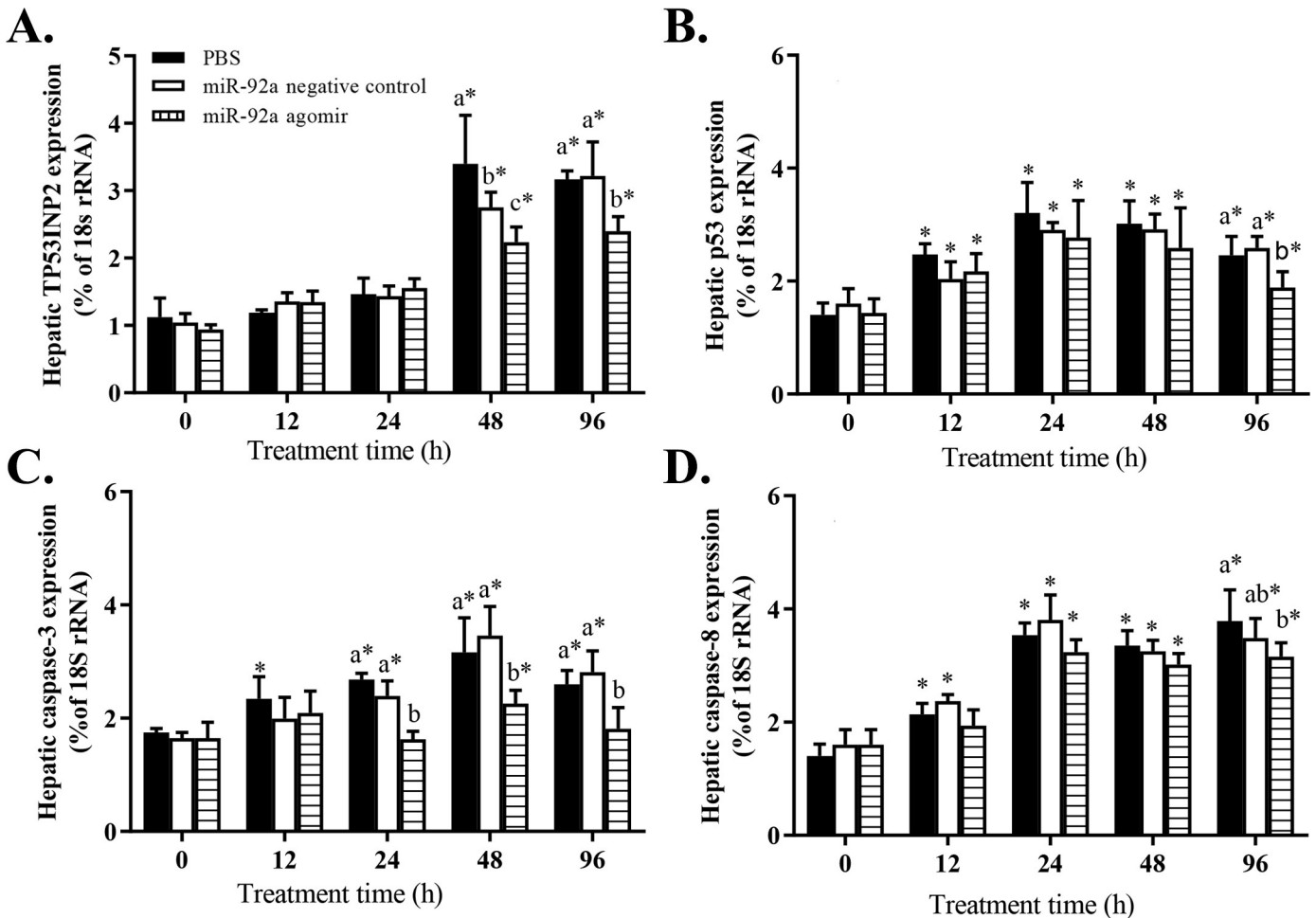

**Fig 7.** Effect of promoting miR-92a expression on transcript levels *p53-inducible nuclear protein 2* (A), *p53* (B), *caspase-3* (C), and *caspase-8* (D) in GIFT under hypoxia stress. At 12 h after injection with miR-92a agomir or PBS (control), juveniles were subjected to hypoxia stress for 96 h. Based on relative expression level in control group at 0 h, relative expression levels of mRNAs in each experimental group were determined by $2^{-\Delta\Delta CT}$ method. * Indicates significant differences between pre- and post- injection in the same experimental group (independent samples t test; $P < 0.05$). Different lowercase letters indicate significant differences among different treatments at each sampling point (Duncan's multiple comparison; $P < 0.05$).

## Discussion

Calcium ions are second messengers of signal transduction and can trigger a variety of cellular and physiological responses, such as muscle contraction, exocytosis, neurotransmitter release, cell proliferation, and apoptosis [8, 22]. In this study, the $Ca^{2+}$ concentration in GIFT hepatocytes increased sharply under hypoxia stress, which may explain the increased proportions of cells in the early and late stages of apoptosis. Previous studies have shown that an increase in the $Ca^{2+}$ concentration induces expression of pro-inflammatory cytokines, thereby activating caspase-9 in the mitochondrial pathway or caspase-8 in the death receptor pathway, via the apoptosis-inducing effects of caspase-3 [23]. However, in hepatocytes of GIFT with knocked-down *CaSR* expression, the decreased CaSR expression levels may have prevented the accumulation of $Ca^{2+}$ under hypoxia stress. This would explain the reduced transcript levels of *p53*, *TP53INP2*, *caspase-3*, and *caspase-8* in hepatocytes with knocked-down *CaSR* expression. Therefore, *CaSR* knockdown helped to maintain the homeostasis of the $Ca^{2+}$ concentration inside and outside the cell, and relieve hypoxia stress-induced apoptosis.

Hypoxia-induced apoptosis is an anoxic adaptation mechanism that eliminates stressed cells (Shimizu et al., 1996). Under hypoxia stress, activation of p53 can induce cell necrosis, apoptosis, or autophagy, all of which are involved in cell death [12, 24]. Previous studies have shown that hypoxia stress leads to increased expression of the *p53* gene and p53 protein in oriental river prawn (*Macrobrachium nipponense*) and in the hepatopancreas and hemocytes of Pacific white shrimp (*Litopenaeus vannamei*), leading to apoptosis [24–26]. Similar results were found in this study. The transcript level of *p53* in each experimental group gradually increased under hypoxia stress. Therefore, GIFT may initiate the p53 signaling pathway through the central axis of hypoxia-miR-92a-CaSR under hypoxia stress. In the control group, the down-regulation of miR-92a and up-regulation of *CaSR* at 48 h and 72 h may have increased cell apoptosis and alleviated stress damage caused by hypoxia. However, activated endogenous expression of miR-92a in GIFT by injection of an miRNA agomir inhibited *CaSR* mRNA expression levels, thereby regulating the apoptosis response. Therefore, decreased *CaSR* expression in the agomir group may have helped to reduce the increase in p53 expression and alleviate apoptosis induced by hypoxia stress. As a key regulatory gene in the p53 regulatory network, *TP53INP2* plays an important role in the apoptosis and invasion of tumor cells. In previous studies, up-regulation of this gene was shown to induce cell apoptosis of glioma cells, breast cancer cells, and osteosarcoma cells [27, 28]. Up-regulation of p53 may result in a significant up-regulation of *TP53INP2*, leading to increased hepatocyte apoptosis and liver damage. However, in our study, *p53* was significantly down-regulated in the liver of GIFT during prolonged exposure to hypoxia stress. Liu et al. [29] suggested that p53 is a universal receptor for environmental stress. Therefore, down-regulation of *p53* and *TP53INP2* may be a self-adaptive mechanism under hypoxia stress in aquatic animals. In another study, after 48 h of hypoxia stress, Pacific white shrimp with silenced *p53* showed enhanced expression of cyclin-dependent kinase 2 (which is involved in cell cycle progression) and decreased caspase-3 expression, and these changes in expression regulated hepatocyte apoptosis and the cell cycle [30].

In our study, the transcript levels of *caspase-3* and *caspase-8* increased significantly in the liver tissues of GIFT under hypoxia stress, suggesting that the caspase signaling pathway may play a role in stress adaptation. Caspase signaling under various stress conditions has been observed in other fish species. For example, cadmium stress was found to induce apoptosis of purse red common carp (*Cyprinus carpio*) hepatocytes, and caspase-3A activity was significantly increased in liver tissues [31]. However, copper stress did not lead to up-regulation of *caspase-3* in Nile tilapia, nor did it increase the proportion of apoptotic cells, suggesting that caspase-3-dependent or caspase-independent apoptotic pathways may not exist in this fish [32]. In our study, up-regulation of miR-92a in the agomir group inhibited the expression of the target gene *CaSR* and reduced the increase in *caspase-3* and *caspase-8* transcript levels under hypoxia stress. This may have prevented apoptosis of GIFT liver cells under hypoxia stress. In another study, down-regulation of miR-532-5p expression in H9c2 cells exposed to hypoxia led to up-regulation of its target gene encoding programmed cell death protein 4, and increased the expression of caspase-3, thereby promoting hypoxia-induced apoptosis of H9c2 cells [33]. Inhibition of miR-9 up-regulated the gene encoding Yes-associated protein 1, promoted cell proliferation, and inhibited apoptosis and caspase-3/7 activity in hypoxic H9c2 cells [34].

Under normal conditions, AST and ALT in fish are mainly located in the liver, and only small amounts are released into the blood. Thus, serum AST and ALT activities are important indicators of normal liver function [35]. In other studies, increased serum AST and ALT activities in GIFT have been detected under high temperature [36] and crowding [37]. In our study, the GIFT liver may have been damaged by hypoxia stress. An increase in the membrane

permeability of liver cells would lead to the release of AST and ALT, resulting in increased serum AST and ALT activities. Inhibition of *CaSR* in the agomir group may have helped to alleviate hepatocyte damage, thereby reducing serum AST and ALT activities.

## Conclusions

This study confirmed the central hypothetic axis of the hypoxia-miR-92a-CaSR-apoptotic phenotype. Up-regulation of miR-92a led to down-regulation of its target gene *CaSR* in GIFT. Stimulation of miR-92a interfered with hypoxia-induced apoptosis in GIFT hepatocytes by targeting *CaSR*, and alleviated liver damage. Our study provides novel insights into the adaptation mechanism of GIFT to hypoxia, and suggests that miR-92a might be a target for relieving stress in farmed fish.

## Supporting information

**S1 Fig. Marker map of CaSR protein expression in GIFT.** CaSR protein (66kD) was detected in GIFT liver sample. The procedures for SDS-PAGE preparation, protein sample electrophoresis, membrane transfer, blocking, and antibody incubation were as described by Qiang et al. [20]. Color was developed using Immobilon Western HRP substrate (Millipore, Billerica, MA, USA). (PDF)

**S2 Fig. Western blot of CaSR expression in GIFT both miR-92a agomir group and PBS group.** CaSR protein (66kD) in miR-92a agomir group (A-D) was gradually weaken after 24 hours but was absent by 48 hours. CaSR protein (66kD) in PBS group (E-H) was used as a control and was present in liver samples. The procedures for SDS-PAGE preparation, protein sample electrophoresis, membrane transfer, blocking, and antibody incubation were as described by Qiang et al. [20]. Color was developed using Immobilon Western HRP substrate (Millipore, Billerica, MA, USA). (PDF)

**S3 Fig. Western blot of GAPDH expression in GIFT both miR-92a agomir group and PBS group.** GAPDH (38 kD) was used as a loading control and was present in GIFT liver samples (A-H). The procedures for SDS-PAGE preparation, protein sample electrophoresis, membrane transfer, blocking, and antibody incubation were as described by Qiang et al. [20]. Color was developed using Immobilon Western HRP substrate (Millipore, Billerica, MA, USA). (PDF)

## Acknowledgments

We thank Jennifer Smith, PhD, from Liwen Bianji, Edanz Group China (http://www.liwenbianji.cn/ac), for editing the English text of a draft of this manuscript.

## Author Contributions

**Data curation:** Jun Qiang.

**Formal analysis:** Yi-Fan Tao.

**Methodology:** Jin-Wen Bao, Jun-Hao Zhu.

**Project administration:** Pao Xu.

**Supervision:** Jun Qiang, Jie He, Pao Xu.

**Validation:** Yi-Fan Tao, Jin-Wen Bao, Jun-Hao Zhu.

**Writing – original draft:** Jun Qiang.

**Writing – review & editing:** Jie He, Pao Xu.

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
