## [Decision Letter · Decision Letter 0]

11 Sep 2020

PONE-D-20-25810

Hypoxia-induced miR-92a regulates p53 signalling pathway and apoptosis by targeting calcium-sensing receptor in Genetically Improved Farmed Tilapia (Oreochromis niloticus)

PLOS ONE

Dear Dr. Qiang,

Thank you for submitting your manuscript to PLOS ONE. After careful consideration, we feel that it has merit but does not fully meet PLOS ONE’s publication criteria as it currently stands. Therefore, we invite you to submit a revised version of the manuscript that addresses the points raised during the review process.

Please respond to all critique, point-by-point. In particular:

- statistical analysis and language may need a checkup by specialists

- all figures (i.e. 2C,D) should be referred to in the main text

We look forward to receiving your revised manuscript.

Kind regards,

Klaus Roemer

Academic Editor

PLOS ONE

Journal Requirements:

2.Thank you for stating the following in the Funding Section of your manuscript:

[The study was supported financially by Central Public-interest Scientific Institution Basal Research Fund, CAFS (NO. 2018HY-XKQ02-01; 2019ZY19; 2019JBFC01).]

 [The author(s) received no specific funding for this work.]

3.PLOS ONE now requires that authors provide the original uncropped and unadjusted images underlying all blot or gel results reported in a submission’s figures or Supporting Information files. This policy and the journal’s other requirements for blot/gel reporting and figure preparation are described in detail at https://journals.plos.org/plosone/s/figures#loc-blot-and-gel-reporting-requirements and https://journals.plos.org/plosone/s/figures#loc-preparing-figures-from-image-files. When you submit your revised manuscript, please ensure that your figures adhere fully to these guidelines and provide the original underlying images for all blot or gel data reported in your submission. See the following link for instructions on providing the original image data: https://journals.plos.org/plosone/s/figures#loc-original-images-for-blots-and-gels.

Reviewers' comments:

Reviewer's Responses to Questions

**Comments to the Author**

1. Is the manuscript technically sound, and do the data support the conclusions?

Reviewer #1: Yes

Reviewer #2: Yes

2. Has the statistical analysis been performed appropriately and rigorously? 

Reviewer #1: I Don't Know

Reviewer #2: No

3. Have the authors made all data underlying the findings in their manuscript fully available?

Reviewer #1: Yes

Reviewer #2: No

4. Is the manuscript presented in an intelligible fashion and written in standard English?

Reviewer #1: No

Reviewer #2: No

5. Review Comments to the Author

Reviewer #1: General comments.

The author studied hypoxia introduces the liver damage by though which the miR-92a and CaSR-apoptosis axis. Then, the author identifies the usefulness of miR-92a to preventing liver dysfunction in fish. The study is exciting and scientifically sound. The author investigated using a cell culture model following the in vivo assay, which is very good. However, there are several missing sentences or insufficient explanations in the text. Usually, this type of careless miss is not suitable for the judgment of the review process, as such, I may not be able to correctly justify the results and may point out with my miss-understand. Please carefully consider and correct the manuscript.

Major comments,

L68-77: Introduce miR-92a in a tumor, but this study did not focus on the oncogenesis. Do you think this part is necessary? I may assume that the author may want to conduct tumor biology and the apoptosis phenomenon, but the author clearly introduces this issue later in the introduction.

L147: Although the author describes that the Liver sample was frozen in liquid nitrogen within 1 h, I think 1 hour is relatively long periods. I am sorry, but I am not familiar with sample collection from fish, is this a typical method for fish? To my knowledge, especially, the mRNA profile in the sample may change during 1 hour. I assume that there is some time lag for sample correcting and frozen in the first sample and last sample. So this means that the author’s experimental condition may not be suitable for downstream analysis. Is this really 1h? If so, the author needs to explain the situation for sample collecting. In addition, the author also collects the liver sample to describe L172; the description is a little different. Did the author use different procedures in each experiment?

In figure 1, the results show protein expression as western blot band signals. However, there is no description of miR-92a, and there is no indication “Figure 2C, 2D” in the text.

L 247: There is (Figure 3B), is this correct? All of Figure 3 shows similar results, isn’t it?

In Figure 5 and L 255-260: The author did not indicate the change of miR-92a by the time; I mean, the author should explain and describe the decreasing of miR-92a by hypoxic stimulation. I think that this is most important for this study hypothesis. So the central hypothetic axis is hypoxia-miR-92a-CaSR-Apsoptotic phenotype, isn’t it? If I miss understanding, I am sorry. The author also needs to clearly discuss this phenomenon in the discussion as well.

Minor comments.

For reference citation, there are many “Qiang et al., 2017”, so I can not clearly evaluate which is which? Besides, the reference citation style is not correct for PLoS One.

Please add the number of the Ethics approval.

L131: It is better to add the sequence of C. elegans in the text for reproducibility by other researchers if necessary.

L139-140: I think that the author needs to show the sequence of miR-92a agomir and negative agomir in text. The author describes the sequence for RNAi experiments.

L158: “normal” means normoxia condition? The author also needs to show how much O2 tension in the control group? I know 20% in most of the cell culture, but still better to describe.

L168: What is section 2.2?

L184: I think I can not find U6 sequence in Qiang 2020, Front. Physiol. It will be better to add.

L212-214: Please describe the exact methods for comparative statistics. Also, if the data did not meet the normal distribution, the author used what statistics methods?

L257-260: Figure A and B may be better to change order.

L261-265: I think it is better to coordinate the order AST and ALT, the author mixture both in text and figure.

Reviewer #2: miR-92a RNAs are important immune molecules involved in regulating cell apoptosis. The research found that miR-92a mediate the immune mechanism of reducing liver cell damage in genetically improved farmed tilapia by targeting CaSR.

My specific comments are as follows:

Question 1:

Line 51-52: “sea cucumber Apostichopus japonicas”, revise to “sea cucumber (Apostichopus japonicas)”; Line 60: “In genetically improved farmed tilapia (GIFT, Oreochromis niloticus)”, revise to “In genetically improved farmed tilapia (GIFT), (Oreochromis niloticus)”, and unify the format about same question in manuscript.

Question 2:

Line 71: “Bcl-2 interacting mediator of cell death (Bim)”, revise to “ Bcl-2 interacting mediator (Bim) of cell death ”.

Question 3:

Line 85: If there is any research on apoptosis mediated by CaSR, please add in the introduction.

Question 4:

Line 103: Please introduce the research on the effects of DO supplementation on the expression of fish apoptosis and the immune response involve proteins.

Question 5:

Line 105: “Photobacterium damselae ssp. (Reis et al., 2007; 2010). ” , revise to: “ Photobacterium damselae ssp. piscicida (Reis et al., 2007; 2010). ”

And the same question,

Line 497-499: “ Reis, M.I. Costa-Ramos, C., do Vale, Ana., dos Santos, N.M.S., 2010. Molecular cloning of sea bass (Dicentrarchus labrax L.) caspase-8 gene and its involvement in Photobacterium damselae ssp. piscicida triggered apoptosis.”, revise to “… involvement in Photobacterium damselae ssp. piscicida …”.

Question 7:

Line 130: “We used Renilla luciferase activity… ”, revise to “We used renilla luciferase activity”.

Question 8:

Line 132: “in the miRBase database”, please add the URL of web site.

Question 9:

Line 162: “and then centrifuged (12000 g, 20 min, 4 °C)”, revise to “ and then centrifuged (4 °C, 12000 g, 20 min) ”; also,

Line 188: “……was added and the mixture was homogenized (15,000 g, 1 min, 4 °C)”, revise to “……was added and the mixture was homogenized (4 °C, 15,000 g, 1 min)”.

Question 10:

Line 167: “……were as described in section 2.2.”, revise to “……were as described in section “Identification of binding sites of miR-92a-CaSR 3’UTR”.”.

Question 11:

Line 216: Please unify the header format in the results.

Question 12:

Line 229: “To analyse the regulatory relationship between miR-92a and its target gene, we injected the miR-92a agomir. negative control (NC) or PBS (control) into the tail vein of GIFT.”, revise to “To analyse the regulatory relationship between miR-92a and its target gene, we injected the miR-92a agomir (negative control (NC) or PBS (control) into the tail vein of GIFT).”

Question 13:

Line 291: “and white shrimp”, revise to “and Pacific white shrimp”.

Question 14:

Line 374 & Line 417: “western blot”, revise to “ Western blot”.

Question 15:

Figures: Please note the significant difference in the figure and who has the difference compared with whom, unify the format of each figure.

Question 16:

According to the journal request, unify the format of reference in the text.

6. PLOS authors have the option to publish the peer review history of their article (what does this mean?). If published, this will include your full peer review and any attached files.

Reviewer #1: No

Reviewer #2: **Yes: **Qinghua Zhang

---

## [Author Response · Author response to Decision Letter 0]

20 Oct 2020

Detailed responses to comments from the editors and reviewers are provided below. Changes are shown in the TrackedCopy manuscript.

Editors:

Comment 1: statistical analysis and language may need a checkup by specialists Response: We have again revised the language and added details of the statistical analyses.

Comment 2: all figures (i.e. 2C,D) should be referred to in the main text

Response: We have ensured that all figures are referred to in the text.

Comment 3: Please ensure that your manuscript meets PLOS ONE's style requirements Response: We have revised the manuscript to ensure that it meets the style requirements of PLoS ONE.

Comment 4: We note that you have provided funding information that is not currently declared in your Funding Statement. However, funding information should not appear in the Acknowledgments section or other areas of your manuscript. We will only publish funding information present in the Funding Statement section of the online submission form

Response: We have revised funding information from the paper.

Comment 5: PLOS ONE now requires that authors provide the original uncropped and unadjusted images underlying all blot or gel results reported in a submission’s figures or Supporting Information files 

Response: We have supplied the original result in the Supporting Information file, and revised the format and file name. Please see S1_raw_images.

Reviewer 1: 

Comment 1: L68-77: Introduce miR-92a in a tumor, but this study did not focus on the oncogenesis. Do you think this part is necessary? I may assume that the author may want to conduct tumor biology and the apoptosis phenomenon, but the author clearly introduces this issue later in the introduction.

Response: We agree with this point, we have rearranged the sentences in the introduction and removed unnecessary information. (Line 68) 

Comment 2: L147: Although the author describes that the Liver sample was frozen in liquid nitrogen within 1 h, I think 1 hour is relatively long periods. I am sorry, but I am not familiar with sample collection from fish, is this a typical method for fish? To my knowledge, especially, the mRNA profile in the sample may change during 1 hour. I assume that there is some time lag for sample correcting and frozen in the first sample and last sample. So this means that the author’s experimental condition may not be suitable for downstream analysis. Is this really 1h? If so, the author needs to explain the situation for sample collecting. In addition, the author also collects the liver sample to describe L172; the description is a little different. Did the author use different procedures in each experiment?

Response: We have revised the incorrect information in our manuscript. During the sampling process, liver was rapidly excised, snap frozen in liquid nitrogen, and then stored at −80 °C until later use [Lines 124-125]. For the whole experiment, sampling was completed within 1 h. After extracting RNA, we further analyzed its integrity (see figure 1 below) and the OD value (OD 260/280 nm), the OD value was 1.9–2.1. We used a Nanodrop 2000 instrument to measure the RNA concentration (800–1000 ng/μL), and the quality of the RNA was confirmed to be sufficient for subsequent analysis. The liver sample collection process described in L172 is the same as that in L147. In our original manuscript, we deleted some of our descriptions to avoid repeating information. We have revised the descriptions of our methods to be more accurate in the revised manuscript [Lines 168-169].

Figure 1. Analysis of completeness of RNA extraction from some samples.

Comment 3: In figure 1, the results show protein expression as western blot band signals. However, there is no description of miR-92a, and there is no indication “Figure 2C, 2D” in the text.

Response: We have added a description of miR-92a in the text of the Results section [Lines 223-225; 227-228]. We have replaced the order of Fig 1 and Fig 2 according to the recommendations of reviewer. Figure 1C and 1D were shown in Line 225-227.

Comment 4: L 247: There is (Figure 3B), is this correct? All of Figure 3 shows similar results, isn’t it?

Response: We have carefully checked the results and figures and confirmed that the results are correct. In this experiment, the normal group (NG, no transfection, no hypoxia) served as the control; that is, the expression level of each gene was set to 1 in the NG samples. The figure shows that hypoxia stress significantly upregulated CaSR (Figure 3A), TP53INP2 (Figure 3B), p53 (Figure 3C), caspase-3 (Figure 3D), and caspase-8 (Figure 3E) in GIFT hepatocytes. However, these genes were not upregulated under hypoxia stress in the CaSR-knockdown line. The experimental results indicated that CaSR may regulate the p53 signaling pathway to interfere with apoptosis in GIFT under hypoxia stress. We also analyzed other genes in related pathways and apoptosis genes in our experiments. Some showed no significant differences or differences that could not be explained. The genes discussed in this paper showed significant differences in their transcript levels among the different treatment groups and/or between hypoxia stress and normal conditions. Thus, these genes can better explain the regulatory mechanism of the hypoxia-miR-92a-CaSR-apoptotic phenotype.

Comment 5: In Figure 5 and L 255-260: The author did not indicate the change of miR-92a by the time; I mean, the author should explain and describe the decreasing of miR-92a by hypoxic stimulation. I think that this is most important for this study hypothesis. So the central hypothetic axis is hypoxia-miR-92a-CaSR-Apoptotic phenotype, isn’t it? If I miss understanding, I am sorry. The author also needs to clearly discuss this phenomenon in the discussion as well.

Response: The reviewer is correct, the central hypothetic axis of this study is the hypoxia-miR-92a-CaSR-apoptotic phenotype. We have added a detailed description in the results [Lines 305-309] and discussion sections [Lines 368-372; 394-397].

Comment 6: For reference citation, there are many “Qiang et al., 2017”, so I can not clearly evaluate which is which? Besides, the reference citation style is not correct for PLoS One.

Response: We have revised the format of citations and references in the paper according to the PLoS ONE style.

Comment 7: Please add the number of the Ethics approval.

Response: The registration number of this experiment is 2019-028. We have added this information to the revised manuscript. [Line 109]

Comment 8: L131: It is better to add the sequence of C. elegans in the text for reproducibility by other researchers if necessary. 

Response: We added the sequence. [Lines 133-134]

Comment 9: L139-140: I think that the author needs to show the sequence of miR-92a agomir and negative agomir in text. The author describes the sequence for RNAi experiments.

Response: We have added the sequences of the miR-92a agomir and negative agomir. [Lines 117–119] 

Comment 10: L158: “normal” means normoxia condition? The author also needs to show how much O2 tension in the control group? I know 20% in most of the cell culture, but still better to describe.

Response: Yes, normal conditions were normoxic conditions. We have added this description to the text. In the experiment, our normally cultured hepatocytes were placed in an Herocell 180 incubator (RADOBIO, Shanghai, China) under normoxic conditions (27 °C, O2 20%, CO2 5%).

Comment 11: L168: What is section 2.2?

Response: We have revised the incorrect description [Lines 162-163]. 

Comment 12: L184: I think I can not find U6 sequence in Qiang 2020, Front. Physiol. It will be better to add.

Response: The U6 gene that we used was from the Takara miRNA SYBR Green qRT-PCR kit and the sequence information is confidential to the company.

Comment 13: L212-214: Please describe the exact methods for comparative statistics. Also, if the data did not meet the normal distribution, the author used what statistics methods?

Response: We have described the statistical analyses in more detail. [Lines 213–216]

Comment 14: L257-260: Figure A and B may be better to change order.

Response: We have revised the order of Figures A and B, and modified the description in the results section accordingly.

Comment 15: L261-265: I think it is better to coordinate the order AST and ALT, the author mixture both in text and figure.

Response: We now describe AST and ALT in the same order in the text as in the figure. 

Reviewer 2

Comment 1: Line 51-52: “sea cucumber Apostichopus japonicas”, revise to “sea cucumber (Apostichopus japonicas)”; Line 60: “In genetically improved farmed tilapia (GIFT, Oreochromis niloticus)”, revise to “In genetically improved farmed tilapia (GIFT), (Oreochromis niloticus)”, and unify the format about same question in manuscript.

Response: We have standardized the style of common and species names in the revised manuscript [Lines 54; 62].

Comment 2: Line 71: “Bcl-2 interacting mediator of cell death (Bim)”, revise to “ Bcl-2 interacting mediator (Bim) of cell death ”.

Response: We deleted this part during the revision of our manuscript.

Comment 3: Line 85: If there is any research on apoptosis mediated by CaSR, please add in the introduction.

Response: We have added information about how CaSR regulates cellular inflammatory responses and apoptosis signaling, as well as supporting references. [Lines 69 -71]

Comment 4: Line 103: Please introduce the research on the effects of DO supplementation on the expression of fish apoptosis and the immune response involve proteins.

Response: We have added more information about the effect of hypoxia stress on fish apoptosis and the immune response, supported by references 12–18. [Lines 88-96]

Comment 5: Line 105: “Photobacterium damselae ssp. (Reis et al., 2007; 2010). ” , revise to: “ Photobacterium damselae ssp. piscicida (Reis et al., 2007; 2010). ”

And the same question,

Line 497-499: “ Reis, M.I. Costa-Ramos, C., do Vale, Ana., dos Santos, N.M.S., 2010. Molecular cloning of sea bass (Dicentrarchus labrax L.) caspase-8 gene and its involvement in Photobacterium damselae ssp. piscicida triggered apoptosis.”, revise to “… involvement in Photobacterium damselae ssp. piscicida …”.

Response: We removed these references during the revision of the manuscript. In the revised manuscript, we have focused on the impact of hypoxia stress on apoptosis in fish.

Comment 6: Line 130: “We used Renilla luciferase activity… ”, revise to “We used renilla luciferase activity”.

Response: Corrected [Line 132]. 

Comment 7: Line 132: “in the miRBase database”, please add the URL of web site.

Response: We have added the URL of the website [Line 135].

Comment 8: Line 162: “and then centrifuged (12000 g, 20 min, 4 °C)”, revise to “ and then centrifuged (4 °C, 12000 g, 20 min) ”; also,

Line 188: “……was added and the mixture was homogenized (15,000 g, 1 min, 4 °C)”, revise to “……was added and the mixture was homogenized (4 °C, 15,000 g, 1 min)”.

Response: We revised the description as you suggested [Lines 154; 186-187].

Comment 9: Line 167: “……were as described in section 2.2.”, revise to “……were as described in section “Identification of binding sites of miR-92a-CaSR 3’UTR”.”

Response: We have revised the text as you suggested [Lines 162-163].

Comment 10: Please unify the header format in the results.

Response: We have used a standard header format in the revised manuscript.

Comment 11: Line 229: “To analyse the regulatory relationship between miR-92a and its target gene, we injected the miR-92a agomir. negative control (NC) or PBS (control) into the tail vein of GIFT.”, revise to “To analyse the regulatory relationship between miR-92a and its target gene, we injected the miR-92a agomir (negative control (NC) or PBS (control) into the tail vein of GIFT).”

Response: We have revised the text as you suggested [Lines 220-221]. 

Comment 12: Line 291: “and white shrimp”, revise to “and Pacific white shrimp”.

Response: Corrected [Line 266].

Comment 13: Line 374 & Line 417: “western blot”, revise to “ Western blot”.

Response: The correct spelling of ‘western blot’ is with a lowercase letter. (Only Southern blot is spelled with an uppercase letter, because it is named after a person.)

Comment 14: Figures: Please note the significant difference in the figure and who has the difference compared with whom, unify the format of each figure.

Response: In the figures, * indicates significant difference between pre- and post- injection in the same experimental group (Independent samples t test; P < 0.05), and different lowercase letters indicate significant differences among different treatments at each sampling point (Duncan’s multiple range test; P < 0.05).

This information is provided in each figure legend.

Comment 15: According to the journal request, unify the format of reference in the text.

Response: We have revised the references in the manuscript.

---

## [Editor Report · Decision Letter 1]

28 Oct 2020

Hypoxia-induced miR-92a regulates p53 signalling pathway and apoptosis by targeting calcium-sensing receptor in Genetically Improved Farmed Tilapia (Oreochromis niloticus)

PONE-D-20-25810R1

Dear Dr. Qiang,

We’re pleased to inform you that your manuscript has been judged scientifically suitable for publication and will be formally accepted for publication once it meets all outstanding technical requirements.

Kind regards,

Klaus Roemer

Academic Editor

PLOS ONE
---

## [Editor Report · Acceptance letter]

4 Nov 2020

PONE-D-20-25810R1 

Hypoxia-induced miR-92a regulates p53 signaling pathway and apoptosis by targeting calcium-sensing receptor in Genetically Improved Farmed Tilapia *(Oreochromis niloticus)*

Dear Dr. Qiang:

I'm pleased to inform you that your manuscript has been deemed suitable for publication in PLOS ONE. Congratulations! Your manuscript is now with our production department. 

Kind regards, 

on behalf of

Dr. Klaus Roemer 

Academic Editor

PLOS ONE